# Defining the Roles of Pyruvate Oxidation, TCA Cycle, and Mannitol Metabolism in Methicillin-Resistant *Staphylococcus aureus* Catheter-Associated Urinary Tract Infection

Santosh Paudel,[a] Sarah Guedry,[a] Chloe L. P. Obernuefemann,[b] Scott J. Hultgren,[b] Jennifer N. Walker,[c,d] Ritwij Kulkarni[a]

[a]Department of Biology, University of Louisiana at Lafayette, Lafayette, Louisiana, USA
[b]Center for Women's Infectious Disease Research, Department of Molecular Microbiology, Washington University School of Medicine, St. Louis, Missouri, USA
[c]Department of Microbiology and Molecular Genetics, McGovern Medical School, University of Texas Health Science Center at Houston, Texas, USA
[d]Department of Epidemiology, Human Genetics, and Environmental Science, School of Public Health, University of Texas Health Science Center at Houston, Texas, USA

**ABSTRACT** Methicillin-resistant *Staphylococcus aureus* (MRSA) is an important cause of complicated urinary tract infection (UTI) associated with the use of indwelling urinary catheters. Previous reports have revealed host and pathogen effectors critical for MRSA uropathogenesis. Here, we sought to determine the significance of specific metabolic pathways during MRSA UTI. First, we identified four mutants from the Nebraska transposon mutant library in the MRSA JE2 background that grew normally in rich medium but displayed significantly reduced growth in pooled human urine (HU). This prompted us to transduce the uropathogenic MRSA 1369 strain with the transposon mutants in *sucD* and *fumC* (tricarboxylic acid [TCA] cycle), *mtlD* (mannitol metabolism), and *lpdA* (pyruvate oxidation). Notably, *sucD*, *fumC*, and *mtlD* were also significantly upregulated in the MRSA 1369 strain upon exposure to HU. Compared to the WT, the MRSA 1369 *lpdA* mutant was significantly defective for (i) growth in HU, and (ii) colonization of the urinary tract and dissemination to the kidneys and the spleen in the mouse model of catheter-associated UTI (CAUTI), which may be attributed to its increased membrane hydrophobicity and higher susceptibility to killing by human blood. In contrast to their counterparts in the JE2 background, the *sucD*, *fumC*, and *mtlD* mutants in the MRSA 1369 background grew normally in HU; however, they displayed significant fitness defects in the CAUTI mouse model. Overall, identification of novel metabolic pathways important for the urinary fitness and survival of MRSA can be used for the development of novel therapeutics.

**IMPORTANCE** While *Staphylococcus aureus* has historically not been considered a uropathogen, *S. aureus* urinary tract infection (UTI) is clinically significant in certain patient populations, including those with chronic indwelling urinary catheters. Moreover, most *S. aureus* strains causing catheter-associated UTI (CAUTI) are methicillin-resistant *S. aureus* (MRSA). MRSA is difficult to treat due to limited treatment options and the potential to deteriorate into life-threatening bacteremia, urosepsis, and shock. In this study, we found that pathways involved in pyruvate oxidation, TCA cycle, and mannitol metabolism are important for MRSA fitness and survival in the urinary tract. Improved understanding of the metabolic needs of MRSA in the urinary tract may help us develop novel inhibitors of MRSA metabolism that can be used to treat MRSA-CAUTI more effectively.

**KEYWORDS** Krebs cycle, MRSA, TCA cycle, mannitol metabolism, metabolism, methicillin-resistant *Staphylococcus aureus*, pyruvate metabolism, urinary tract infection

Historically, *Staphylococcus aureus* was considered an atypical uropathogen; however, recent reports highlight its clinical significance in the urinary tract (UT) (1–3). These reports indicate that *S. aureus* is an increasing cause of asymptomatic bacteriuria and complicated urinary tract infections (UTIs), primarily in the elderly, those with recent hospitalization,

Address correspondence to Ritwij Kulkarni, ritwij.kulkarni@louisiana.edu.

The authors declare no conflict of interest.

and individuals with indwelling urinary catheters (1, 2, 4–7). Moreover, *S. aureus* UTI has become a clinical concern over the last 2 decades due to the increasing incidence of methicillin-resistant *S. aureus* (MRSA) detected in human urine (HU), indicating that the UT can be a reservoir for drug resistance (6, 8, 9). Additionally, *S. aureus* UT colonization is a known precursor for life-threatening invasive infections such as bacteremia, urosepsis, and shock (6, 9–11). The role of urinary catheterization in predisposed individuals to MRSA UT colonization has also been validated in the C57BL/6 mouse model of catheter-associated UTI (CAUTI) (12). In our previous study, catheterized mice showed 300-fold-higher MRSA bladder burdens at 1 day postinfection (dpi) than their noncatheterized counterparts, and MRSA persistence was marked by significantly higher bacterial burden in the bladder and kidneys of catheterized mice at 14 dpi (12).

The UT is a moderately oxygenated microenvironment containing urine, which is a high-osmolarity, iron-limiting, dilute mixture of amino acids and peptides (13). The successful colonization of the UT is contingent on the ability of uropathogens to transport amino acids and peptides across the membrane and catabolize them via the tricarboxylic acid (TCA) cycle and gluconeogenesis. Indeed, mutants of uropathogenic *Escherichia coli* (UPEC) ablated for peptide transport, as well as mutants of UPEC and *Proteus mirabilis* deficient for TCA cycle and gluconeogenesis, exhibited significantly reduced fitness in the mouse model of ascending UTI (14, 15). In our previous study, we used RNA sequencing to show that 2-h-long exposure of uropathogenic MRSA 1369 to HU induced the expression of oligopeptide transporters (*oppBCDFAA*) and enzymes catalyzing the TCA cycle and gluconeogenesis (16). In the current study, we provide evidence to further support a central role of amino acid catabolism in MRSA uropathogenesis. Our screen of the *S. aureus* Nebraska transposon mutant library (NTML; JE2 strain background) identified mutants that showed normal (wild type [WT]-like) growth in nutrient-rich brain heart infusion (BHI) but displayed a growth defect in HU. These mutants include genes with known roles in pyruvate oxidation, mannitol metabolism, and TCA cycle. Together, the previously published RNA sequencing data and the urine growth analysis from the current study identified four genes with the greatest potential to play a role in MRSA CAUTI pathogenesis (16). Thus, we selected the uropathogenic MRSA 1369 background strain and generated mutants defective in (i) pyruvate oxidation enzyme dihydrolipoamide dehydrogenase (*lpdA*, SAUSA300_0996), (ii) TCA cycle enzyme fumarase (*fumC*, SAUSA300_1801), (iii) succinyl CoA synthase (*sucD*, SAUSA300_1139), or (iv) mannitol metabolism enzyme mannitol-1-phosphate dehydrogenase, also known as M1PDH (*mtlD*, SAUSA300_2108). MRSA 1369 *sucD* and *fumC* mutants did not show delayed *in vitro* growth in HU, although both *sucD* and *fumC* mutants were at a significant competitive disadvantage against WT in an *in vivo* mouse model of CAUTI. In contrast, both *lpdA* and *mtlD* mutants in the MRSA 1369 strain background were significantly defective for *in vitro* growth in HU compared to the WT. Additionally, both *lpdA* and *mtlD* mutants were at a distinct disadvantage compared to the WT in an *in vivo* mouse model of CAUTI. Furthermore, the *lpdA* mutant was also defective in dissemination to the kidneys as well as the spleen. The poor *in vivo* survival of these mutants may be attributed to the higher hydrophobicity and increased susceptibility to killing by whole human blood observed in the *lpdA* mutant and to the higher susceptibility to hydrogen peroxide and whole blood observed in the *mtlD* mutant. These findings support previous reports that *S. aureus* dihydrolipoamide dehydrogenase (LpdA) is involved in the catalysis of the pyruvate oxidation step bridging glycolysis with TCA cycle (17), the regulation of staphylococcal membrane fluidity (18), and the production of staphylococcal protein A (19), while M1PDH (MtlD) is involved in mannitol metabolism and protection from innate antimicrobial effectors such as hydrogen peroxide and linoleic acid (20).

In summary, this work improves our knowledge of how MRSA may utilize specific metabolic pathways for survival and proliferation in the host UT. Our observations may also be applied toward the development of novel therapeutics against MRSA UTI, a clinically relevant endeavor in the face of an ever-shrinking repertoire of effective antibiotics and an expanding pool of emergent drug-resistant strains of *S. aureus*.

**TABLE 1** Comparison of select NTML strains for *in vitro* growth ratios in HU with corresponding differential gene expression values for MRSA 1369 exposed *in vitro* to HU for 2 h

| NTML designation | KEGG locus tag | Gene | Growth ratio at: | | Log$_2$(FC)[a] |
|---|---|---|---|---|---|
| | | | 2 h | 4 h | |
| NE427 | SAUSA300_1801 | *fumC* | 0.41 | 0.49 | 2.69[b] |
| NE1263 | SAUSA300_2108 | *mtlD* | 0.42 | 0.48 | 2.88[b] |
| NE1610 | SAUSA300_0996 | *lpdA* | 0.59 | 0.43 | 0.66 |
| NE1770 | SAUSA300_1139 | *sucC* | 0.36 | 0.49 | 4.09[b] |

[a]Log$_2$(FC) refers to average log$_2$(fold change) of normalized counts for triplicate samples, MRSA 1369 exposed for 2 h *in vitro* to HU compared to that in nutrient-rich TS broth.
[b]Adjusted *P* value of <0.05. The Log$_2$(FC) data are adapted from reference 16.

## RESULTS

**Genome-wide screen for MRSA genes required for growth in HU.** We screened MRSA JE2 and the corresponding 1,920 transposon mutants from the Nebraska transposon mutant library (NTML) for growth in HU by measuring optical density at 600 nm ($OD_{600}$) overnight. The growth ratios [$OD_{600}$(mutant)/$OD_{600}$(WT)] at 1, 2, 4, and 22 h are presented as a heatmap in Fig. S1 in the supplemental material and show mutants in the JE2 background that display significant growth defects in HU, defined as a growth ratio of <0.5. Of these mutants, the vast majority were also defective for growth in BHI (data not shown), indicating a general growth defect. Only six mutants displayed an HU-specific growth defect. Of these mutants, NE427, NE1263, and NE1770, targeting metabolic genes *fumC* (SAUSA300_1801), *mtlD* (SAUSA300_2108), and *sucD* (SAUSA300_1770), respectively, were selected for further analysis, as the expression of *fumC*, *sucD*, and *mtlD* was significantly upregulated following 2 h exposure to HU in our previously published MRSA 1369 transcriptome data (16). The growth ratio for NE1610 targeting *lpdA* (SAUSA300_0996) was 0.59 at 2 h; the *lpdA* expression was also not altered in MRSA 1369 following 2 h exposure to HU (16). However, the NE1610 mutant displayed growth defects in HU at both 4 and 22 h with growth ratios of 0.43 and 0.48, respectively (Fig. S1). Hence, *lpdA* was also selected for further analysis. While NE751 also showed a significant defect for growth in HU at 22 h (growth ratio = 0.14), it was not selected for further analysis because (i) the mutant is also defective for growth in BHI, and (ii) the expression of its target SAUSA300_1693, annotated as a hypothetical protein, is not significantly affected (log$_2$(FC) = 0.46, $P_{adj}$ = 0.27) in MRSA 1369 exposed to HU for 2 h (16). In Table 1, we have presented 2-h and 4-h growth ratios for NE427, NE1263, NE1610, and NE1770 mutants and the differential gene expression values for their respective target genes *fumC*, *mtlD*, *lpdA*, and *sucD*. Next, we used phage transduction to move the corresponding transposon mutants into the MRSA 1369 strain, which is a clinical uropathogenic isolate.

**WT and mutant MRSA 1369 *in vitro* growth in HU and BHI.** We compared the growth of MRSA 1369 WT and mutants in *fumC*, *sucD*, *mtlD*, and *lpdA* in HU and nutrient-rich BHI (Fig. 1). None of the mutants were defective for growth in the nutrient-rich BHI in comparison to the WT strains, as shown by similar growth kinetics (Fig. 1A) and the similar doubling time of ~20 min (Fig. 1B). Additionally, the MRSA 1369 mutants defective in *sucD* ($9.1 \times 10^7$ CFU/mL), *fumC* ($9.6 \times 10^7$ CFU/mL), and *mtlD* ($5.3 \times 10^7$ CFU/mL) displayed similar CFU as the WT ($1.2 \times 0^8$ CFU/mL) after 24 h growth in HU (Fig. 1C). Furthermore, the average doubling times of MRSA 1369 mutants defective in *sucD* ($52 \pm 15$ min), *fumC* ($46 \pm 17$ min), and *mtlD* ($61 \pm 25$ min) were also not significantly different from that of WT ($46 \pm 12$ min) (Fig. 1D). In contrast, the *lpdA* mutant showed 4-fold-lower CFU ($2.8 \times 10^7$ CFU/mL) than the WT after 24 h growth in HU (Fig. 1C) and 5-fold-higher doubling time ($224 \pm 37$ min) than the WT (Fig. 1D). These results suggest that the TCA cycle and mannitol metabolism are not required for growth in HU, while pyruvate oxidation plays an important role in the growth and survival of MRSA 1369 in HU. A schematic of metabolic pathways catalyzed by FumC, SucD, MtlD, and LpdA is shown in Fig. 2.

Next, we compared the growth of the WT and *lpdA* mutant strains in HU supplemented with glucose or with citrate (Fig. S2). The *lpdA* mutant growth in HU with glucose (Fig. S2A)

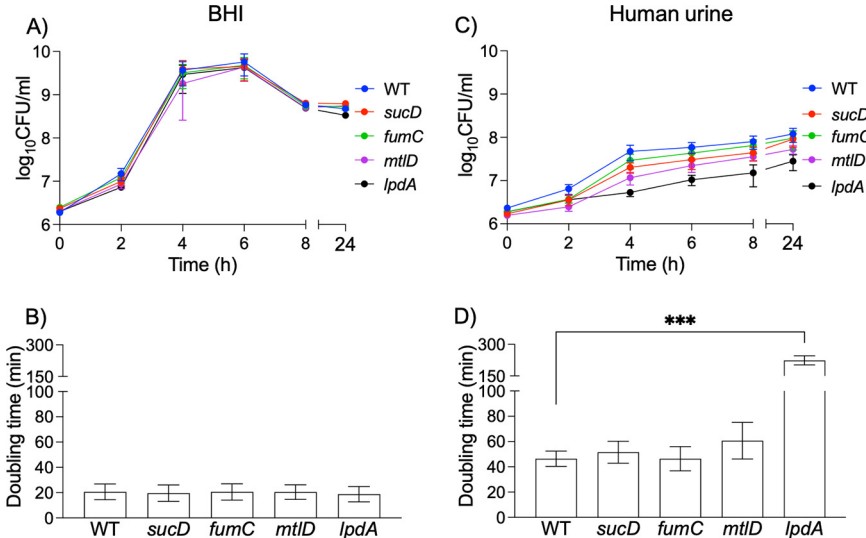

**FIG 1** Growth curves of MRSA 1369 WT and select mutants. MRSA 1369 WT and mutant strains were grown at 37°C, static, for 24 h in either nutrient-rich BHI (A and C) or pooled human urine (B and D), and CFU per milliliter were determined by dilution plating on TSA at specific time points. The growth of WT and mutant MRSA strains are presented as growth curves showing average CFU per milliliter ± standard error of the mean (A and B) and as average doubling time ± standard error of mean (C and D). The data are from 2 biological replicates for BHI and from 3 to 4 biological replicates for human urine; each biological replicate had 2 technical replicates. The doubling time for each mutant was compared with the WT using unpaired *t* test. ***, $P \leq 0.001$ compared to WT control.

or in HU with citrate (Fig. S2C) was affected up to 24-h time point, similar to what was observed in HU alone. However, at the 30-h time point, the *lpdA* mutant showed ∼3-fold-higher CFU/mL in HU with glucose ($3.2 \times 10^7$ CFU/mL, not significant) (Fig. S2B) and ∼2-fold-higher CFU/mL in HU with citrate ($2.1 \times 10^7$ CFU/mL, $P = 0.054$, unpaired *t* test) (Fig. S2D) compared to HU alone ($9.4 \times 10^6$ CFU/mL). These results suggest that the lower growth of *lpdA* mutant in HU can be rescued modestly but statistically insignificantly by

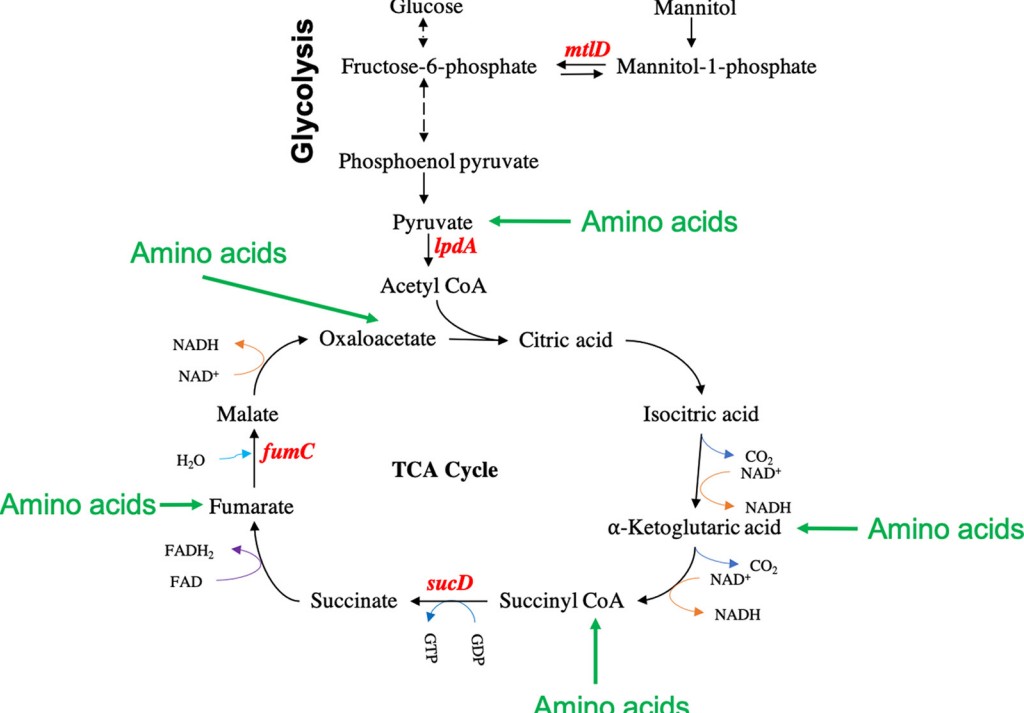

**FIG 2** Schematic showing the steps in metabolic pathways catalyzed by *fumC*, *sucD*, *mtlD*, and *lpdA*.

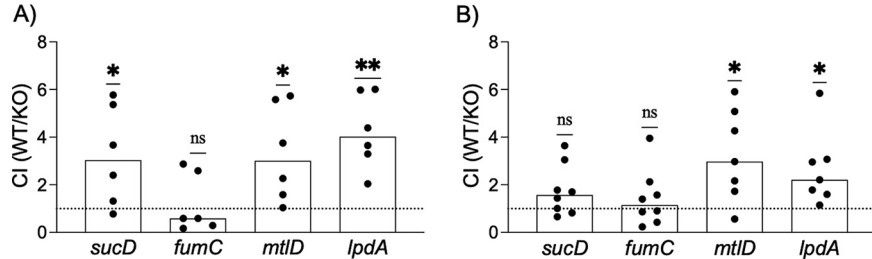

**FIG 3** Competition between WT and mutant MRSA 1369 strains for *in vitro* growth in pooled human urine (HU). WT MRSA 1369 was separately cocultivated with each gene mutant at a 1:1 ratio in pooled HU at 37°C, static. The inoculum CFU as well as the CFU recovered at 4 h and 24 h were enumerated by dilution plating. The competitive index (CI) for each mutant strain at 4 h (A) and 24 h (B) was calculated as CI = (WT recovered/mutant recovered)/(WT inoculum/mutant inoculum). CI values are shown as scatterplots with each point representing a technical replicate and median as the measure of central tendency. A CI of 1, shown as a dotted line, represents that WT and knockout (KO) are equally competitive, a CI of <1 denotes that the KO has a competitive advantage over WT, and a CI of >1 indicates that the WT has a competitive advantage over the KO. Data from 3 or 4 biological replicates, each with 2 technical replicates, are shown. Data were compared against a theoretical median of 1 using one sample *t* test and Wilcoxon test. **, $P \leq 0.01$; *, $P \leq 0.05$.

supplementation with glucose, an alternative carbon source, or citrate, which is downstream of the metabolic step catalyzed by LpdA.

**In vitro competition between WT and mutant MRSA 1369 in HU.** Next, we set up *in vitro* HU cocultures containing equal numbers of MRSA 1369 WT and each of the mutants. The direct competition for amino acids and peptides in the HU was expected to make the urinary fitness advantage of the WT over the mutants more discernible. We enumerated WT and mutant CFU at 0 (inoculum), 4, and 24 h of coculture and calculated the competitive index (CI) for each mutant strain at 4 h (Fig. 3A) and 24 h (Fig. 3B). Similar to the results from the growth curve, the *fumC* mutant was as fit as WT for growth in HU at 4 and 24 h, while the *lpdA* mutant displayed a significant competitive disadvantage compared to WT for growth in HU at both the 4- and 24-h time points. In contrast to the growth curve results, the *mtlD* mutant displayed a significant competitive disadvantage against the WT at 4 and 24 h, while the *sucD* mutant displayed a competitive disadvantage at 4 h, although by 24 h, it appeared to be competitively equal with WT for the growth in HU.

**WT and mutant MRSA 1369 *in vivo* fitness in the mouse model of CAUTI.** To assess the role of metabolism and amino acid synthesis during infection, we used *in vivo* competition experiments in the murine model of CAUTI. We coinfected 8- to 10-week-old female C57BL/6 mice with equal numbers of MRSA 1369 WT and individual mutant strains (1:1 inoculum of WT and mutant). At 24 hours postinfection (hpi), the bacterial burden in the bladder, kidneys, spleen, and catheter was determined by dilution plating. The CFU per milliliter of WT and mutant strains in each mouse and the calculated CI values are presented as scatterplots with median as the central tendency in Fig. 4. Both TCA cycle enzyme mutants defective in *sucD* (Fig. 4A) and *fumC* (Fig. 4C) were less fit for UT colonization, as indicated by significantly higher CI values in the bladder (median CI for *sucD*, 5.6; median CI for *fumC*, 1.8) and the kidneys (median CI for *sucD*, 10; median CI for *fumC*, 3.4). The *sucD* mutant was less fit in colonizing the catheter implant than the WT (median CI, 5.3), while the *fumC* mutant (median CI, 0.8) was not (Fig. 4A and C). Additionally, kidney infection was detected in 73% (8/11) of WT and *sucD*-coinfected mice and 64% (7/11) of WT and *fumC*-coinfected mice. Consistent with the *in vitro* competition data, the *mtlD* mutant was also less fit for colonizing the UT (median CIs, bladder, 3.8; catheter implant, 3.3; and kidneys, 3.5); kidney infection was detected in 75% (6/8) of mice coinfected with WT and *mtlD* (Fig. 4E). Additionally, consistent with the *in vitro* growth and competition data, the *lpdA* mutant was less fit in colonizing the UT, as indicated by significantly higher CI values than the WT for the bladder (median CI, 15.5) and the catheter implant (median CI, 11.4). Notably, only 25% (2/8) of WT and *lpdA*-coinfected mice had detectable kidney infection (Fig. 4G and H). The CFU per milliliter values for WT and mutant strains correlated with the CI; for example, the median CFU per milliliter for mutant was significantly lower than that for WT, where the median CI was >1. Taken together, the results from *in vitro*

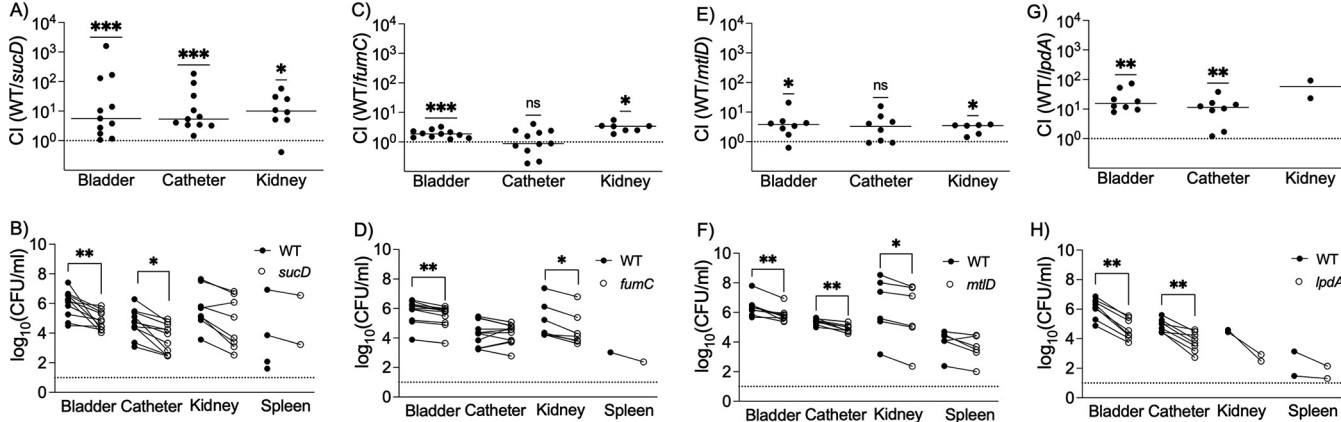

**FIG 4** *In vivo* competition examining the fitness of MRSA 1369 mutants compared to the WT. C57BL/6 female mice were catheterized and then inoculated transurethrally with a 1:1 mixture of MRSA 1369 WT and either *sucD* (A and B), *fumC* (C and D), *mtlD* (E and F), or *lpdA* (G and H) mutants. At 24 hpi, WT and mutant CFU burden in bladder, kidneys, and catheter were determined as shown in panels B, D, F, and H. Competitive index (CI) values for the urinary bladder, kidneys, and catheter from each mouse are presented as scatter diagrams with median as the measure of central tendency as shown in panels A, C, E, and G. A CI of 1, shown as a dotted line, represents that WT and mutant are equally competitive, a CI of <1 denotes that the mutant has a competitive advantage over WT, and a CI of >1 indicates that the WT has a competitive advantage over mutant. The CI data were compared against a theoretical median of 1 using one sample *t* test and Wilcoxon test, and the organ burden data were compared using Wilcoxon matched-pairs signed-rank test; ***, $P \leq 0.001$; **, $P \leq 0.01$; *, $P \leq 0.05$; ns, not significant.

growth in HU and *in vivo* competition in a mouse model of CAUTI indicate that *mtlD* and *lpdA* mutants consistently display fitness defects.

**Uropathogenesis of *lpdA* during monococulture CAUTI.** The *lpdA* mutant not only displayed a competitive disadvantage against the WT in both *in vitro* and *in vivo* experiments but was also defective in dissemination to the kidneys. This prompted us to further examine the uropathogenesis of the *lpdA* mutant via monococulture infection using the mouse model of CAUTI. The *lpdA* mutant displayed significantly lower CFU in the bladder (*lpdA* median, 55,000 CFU/mL; WT, $4.6 \times 10^6$ CFU/mL; $P = 0.0002$), the catheter implant (*lpdA* median, 3,150 CFU/mL; WT, $2.8 \times 10^5$ CFU/mL; $P = 0.0012$), and the kidney (*lpdA* < limit of detection [LOD]; WT, $4.4 \times 10^5$ CFU/mL; $P = 0.037$) than in WT-infected mice (Fig. 5). Notably, the *lpdA* mutant was also defective in dissemination to the kidneys (WT, 1/8, and *lpdA*, 5/8 below LOD) and the spleen (WT, 4/8, and *lpdA*, 8/8 below LOD). Overall, these results revealed that LpdA activity contributes to the colonization of the lower UT as well as the dissemination to the kidneys and the spleen in a mouse model of CAUTI.

**Survival of *mtlD* and *lpdA* mutants in the presence of H₂O₂.** To investigate the potential mechanisms by which LpdA and MtlD enzymes may promote infection, we examined *mtlD* and *lpdA* mutants for survival in the presence of $H_2O_2$, a bactericidal product of activated macrophages and neutrophils. In nutrient-rich BHI, both *mtlD* and *lpdA* mutants were significantly more susceptible to $H_2O_2$ at 4 h (Fig. 6A). Notably, only the *mtlD* mutant was significantly more sensitive to $H_2O_2$ at 4 h in HU (Fig. 6B). These results suggest that under nutrient-rich conditions, regulation of *S. aureus* reactive oxygen species (ROS) stress

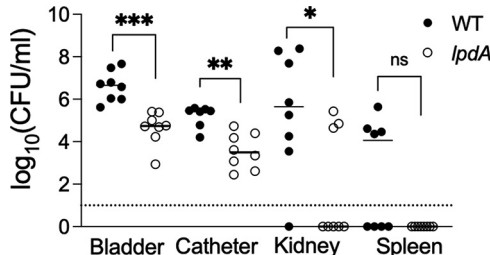

**FIG 5** Urinary pathogenesis of *lpdA* in an *in vivo* mouse model of CAUTI. C57BL/6 female mice were inoculated with WT or *lpdA*. At 24 hpi, WT and Δ*lpdA* CFU in the spleen, bladder, kidneys, and catheter were determined. The data are presented as scatter diagrams showing organ burden from an individual mouse and median as the central tendency. The dotted line represents a limit of detection (LOD) of 10 CFU/mL. Data from 8 mice per group from ≥2 biological replicates were compared using Mann-Whitney U test; ***, $P \leq 0.001$; **, $P \leq 0.01$; *, $P \leq 0.05$; ns, not significant.

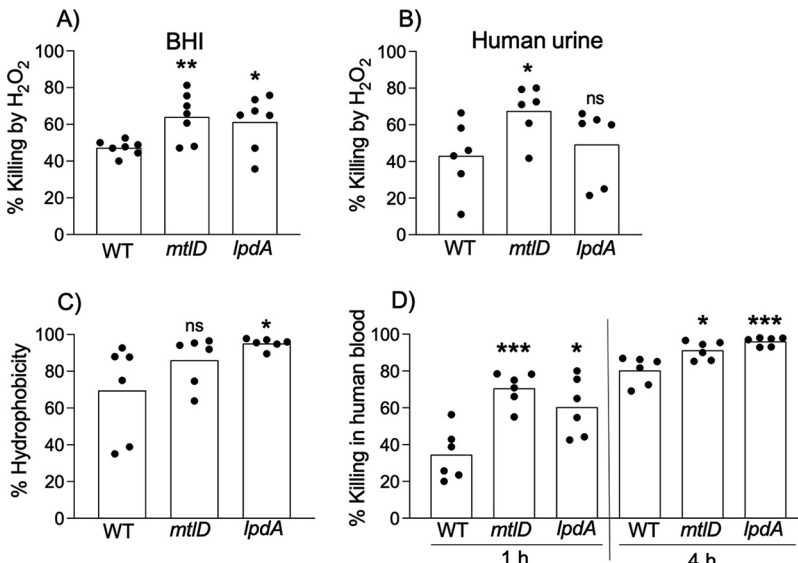

**FIG 6** *mtlD* and *lpdA* susceptibility to $H_2O_2$, cell surface hydrophobicity, and *in vitro* killing by whole blood. WT, *mtlD*, and *lpdA* were exposed to 15 mM $H_2O_2$ for 4 h in either BHI (A) or pooled human urine (HU) (B). The inoculum (0 h) and surviving CFU after 4-h-long exposure to $H_2O_2$ were determined. Additionally, following 2 h exposure to urine, WT and *lpdA* were compared for cell surface hydrophobicity (C) and percent killing in human blood (D). Scatterplots show individual technical replicates from 3 biological replicates with histograms representing the average. The data for each mutant were compared with the WT (at the 1-h or 4-h time point for percent killing in human blood) using unpaired *t* test; **, $P \leq 0.01$; *, $P \leq 0.05$; ns, not significant.

response occurs via either MtlD-mediated replenishment of intracellular mannitol reserves (21) or LpdA-mediated alterations in membrane fluidity (18). However, regulation of *S. aureus* ROS stress response within the UT environment is primarily affected by MtlD.

**Analysis of *lpdA* and *mtlD* mutants' cell surface hydrophobicity.** The contribution of LpdA and MtlD to *in vivo* colonization led us to further assess their virulence contributions. First, we assessed the surface hydrophobicity of the *lpdA* mutant, as LpdA activity can modulate surface hydrophobicity and membrane fluidity through the regulation of branched-chain fatty acid (BCFA) synthesis (22–24). Using microbial adhesion to hydrocarbon assay (MATH) protocol, we compared WT and *lpdA* mutants grown in HU for 2 h for their ability to adhere to the hydrocarbon hexadecane. We observed a significant increase in the hydrophobicity of the *lpdA* mutant compared to the WT (Fig. 6C). The hydrophobicity of the *mtlD* mutant was not significantly affected, as expected since MtlD does not have a known role in membrane fluidity or hydrophobicity (Fig. 6C). These results suggest that lack of LpdA may increase the hydrophobicity of the *lpdA* mutant.

**Survival of *lpdA* and *mtlD* mutants in human blood.** MRSA 1369 typically disseminates to the blood and spleen in ~50% of infected mice at 24 hpi during CAUTI (12). Consistently, in this study, 4/8 (50%) WT-infected mice displayed high CFU in the spleen (Fig. 5). In contrast, the absence of CFU in the spleen in any of the mice singly infected with the *lpdA* mutant at 24 hpi (Fig. 5) suggests that LpdA may be important for survival in blood. Indeed, compared to the WT, the *lpdA* mutant was significantly more susceptible to killing in whole human blood at both 1 h and 4 h following growth in urine (Fig. 6D). Furthermore, the *mtlD* mutant was also more susceptible to killing in whole human blood at both 1 h and 4 h following growth in urine (Fig. 6D). These data suggest that LpdA and MtlD may provide resistance to stress encountered within the UT and during dissemination during CAUTI.

## DISCUSSION

First detected in hospitalized patients in the 1960s, MRSA has spread in the community for the last 3 decades as a commensal colonizing the human skin, nasopharynx, and lower digestive tract (25). MRSA colonization significantly increases the risk of infections ranging from moderately severe skin infections to potentially fatal pneumonia and sepsis (26). In addition, MRSA has also emerged as an important etiology of hospital-acquired UTIs, primarily

arising from the contamination of indwelling urinary catheters (27). Studies using the mouse model of ascending UTI have established that MRSA can colonize murine UTs either in the presence (complicated UTI) or the absence (uncomplicated UTI) of a catheter, and specific virulence traits central to MRSA uropathogenesis have been identified (12, 28). Notably, the contribution of MRSA metabolism to its survival and fitness in the UT has not been deciphered despite the known role of amino acid utilization via the TCA cycle and gluconeogenesis in the *in vivo* urinary fitness of both UPEC and *P. mirabilis* (13–15). Specifically, FumC, the fumarase enzyme, of the TCA cycle has been reported to be required for the colonization of the UT by UPEC and *P. mirabilis* (15). Furthermore, FumC contributes to respiratory survival and persistence of *S. aureus* (29). Interestingly, UT colonization by *P. mirabilis* also requires the activity of glycolysis and Entner-Doudoroff pathways catabolizing glucose into pyruvate, while these glucose utilization pathways are dispensable for the *in vivo* fitness of UPEC (14). The essentiality of the TCA cycle and gluconeogenesis is unsurprising, as the UT in nondiabetic humans is a nutrient-poor, carbohydrate-lacking microenvironment where short peptides and amino acids in urine are the principal carbon sources available to the colonizing uropathogens. We recently reported that exposing MRSA 1369 to HU induces the expression of genes encoding oligopeptide import systems and TCA cycle enzymes while suppressing the expression of glycolysis genes (16). Overall, these observations indicate that different uropathogens rely on divergent central carbon metabolism pathways for successful colonization of the UT.

To further investigate the role of metabolism in MRSA uropathogenesis, we screened the MRSA NTML in the JE2 background and identified mutants defective in *sucD*, *fumC*, *mtlD*, and *lpdA* that were defective for growth in HU but grew normally in nutrient-rich BHI. Notably, we previously reported that in the uropathogenic MRSA 1369 strain, the exposure to HU upregulates the expression of the TCA cycle genes *sucD* (encodes succinyl-CoA synthase) and *fumC* (iron-dependent fumarase) and the mannitol metabolism gene *mltD* (M1PDH); HU exposure did not alter the expression of *lpdA* encoding the dihydrolipoamide dehydrogenase enzyme, catalyzing pyruvate oxidation (16). Here, we observed that in the MRSA 1369 background compared to the WT, *sucD*, *fumC*, and *mtlD* strains were not significantly defective for growth in HU. This surprising deviation from the observations made in the JE2 background mutants may be attributed to the differences in the regulation of metabolic genes or carriage of other genes that differ between MRSA JE2, a skin isolate, and MRSA 1369, a uropathogenic isolate, as the tested genes share 99.99% nucleotide sequence homology between these two strains (30, 31).

MRSA 1369 *sucD* and *fumC* mutants also did not display a competitive disadvantage against WT during *in vitro* growth in HU at 24 h, although the *sucD* mutant showed a median CI of 3.03 ($P = 0.013$) at 4 h. Succinyl-CoA synthase (SucD) regulates the substrate level phosphorylation step in the TCA cycle by catalyzing the hydrolysis of succinyl CoA into succinate and GTP (32), while iron-independent fumarase (FumC) catalyzes conversion of fumarate to malate in the oxidative TCA cycle in *E. coli* (22, 33). The biochemical pathway used by the *sucD* and *fumC* mutants for growth in HU is unclear, as *S. aureus* lacks genes encoding isocitrate lyase and malate synthase, which can bypass the TCA cycle via glyoxylate in the absence of *sucD* and *fumC* (23). Interestingly, however, the *sucD* and *fumC* mutants were at a significant competitive disadvantage against WT in an *in vivo* mouse model of CAUTI, thus confirming the requirement of a functional TCA cycle for MRSA in urinary pathogenesis.

M1PDH encoded by *mtlD* catalyzes the reversible conversion of mannitol-1-phosphate to fructose-6-phosphate, which is the first key step in mannitol fermentation, a distinguishing characteristic of pathogenic *S. aureus*. Fructose-6-phosphate can then be catabolized to pyruvate by Embden-Meyerhoff or glycolytic pathways. While humans do not produce mannitol and the baseline urine mannitol concentration is very low (24), consumption of mannitol sweeteners can raise urine mannitol levels to 14 mg/mL (34). Thus, even though urine mannitol can be converted by MRSA MtlD to fructose-6-phophate for further processing through the energy metabolism pathways, the physiological relevance of this pathway during UTI would depend on the dietary exposure to mannitol sweeteners. Alternatively,

$NADH^+$-dependent M1DPH activity converts fructose-6-phosphate to mannitol-1-phosphate, which can then be dephosphorylated to replenish the intracellular reserves of mannitol that are essential for maintaining cellular redox and osmotic potential in MRSA. Indeed, it has been previously reported that the *S. aureus mtlD* mutant was more sensitive to linoleic acid, an innate immune antimicrobial effector on the skin and hydrogen peroxide (20), and was significantly defective in a mouse model of systemic infection (21). We observed the *mtlD* mutant to be at a significant competitive disadvantage against WT for *in vitro* growth in HU as well as for *in vivo* UT colonization, which may be attributed to the increased sensitivity of *mtlD* mutant to $H_2O_2$ and whole blood *in vitro*.

While our previous study indicates that the gene expression of *lpdA* was not upregulated by exposure to HU (16), this mutant was severely defective for growth in HU. Notably, in addition to dihydrolipoamide dehydrogenase (encoded by *lpdA*, SAUSA300_0996), the *S. aureus* pyruvate dehydrogenase (PDH) complex is made of three additional catalytic peptides, PDH E1$\alpha$ (*pdhA*, SAUSA300_0993), PDH E1$\beta$ (*pdhB*, SAUSA300_0994), and dihydrolipoamide acetyltransferase (SAUSA300_0995) (17). The PDH complex bridges glycolysis with TCA cycle by catalyzing oxidative decarboxylation of pyruvate to acetyl-CoA and shares homology with the branched-chain $\alpha$-keto acid dehydrogenase (BKD) complex. Acetyl-CoA is also a precursor for the synthesis of fatty acids, the principal determinants of staphylococcal cell membrane fluidity (35). Our previous report indicating that a short (2-h-long) exposure to HU does not significantly alter the transcription of the *pdh* gene cluster (*pdhA*, *pdhB*, SAUSA300_0995, and *lpdA*) suggests that at least at early time points, pyruvate oxidation is not needed for MRSA growth in a glucose-poor growth medium such as healthy HU (16). Here, we report that the *lpdA* mutant (i) grew significantly slower than the WT MRSA 1369 and JE2 strains in HU from healthy volunteers, (ii) was outcompeted *in vitro* during growth in healthy HU, (iii) was severely defective (median CI > 10) compared with the WT in an *in vivo* competition mouse model of CAUTI (note that *fumC*, *sucD*, and *mtlD* mutants showed a CI of <5), (iv) was defective in dissemination to mouse kidneys and spleen, and (v) displayed significantly increased cell surface hydrophobicity and susceptibility to killing in blood following a 2-h-long *in vitro* exposure to HU. Together, these data suggest that the reduced fitness of *lpdA* mutant within the UT may be due to LpdA's role in metabolism, which would affect the bacterial survival in HU or due to changes in the cell surface of *lpdA* mutant, resulting in the increased hydrophobicity and higher sensitivity to killing in whole blood. The increased sensitivity to whole-blood killing may also explain why the *lpdA* mutant was not recovered from the splenic homogenates of the mouse model of CAUTI at 24 hpi.

It has been reported that *S. aureus pdhA* activity can be stimulated by osmotic stress (36) and the JE2 *pdhA*::Tn mutant (NE1724) has a higher BCFA content, resulting in increased membrane fluidity (18). Notably, this transposon mutant (NE1724) is defective in all four subunits of PDH, as targeting *pdhA*, which is the first gene in the *pdh* gene cluster, likely affects the expression of the three downstream genes. Furthermore, previous reports indicate NE1724 is defective for growth in rich media (37), which was supported by the results from our NTML screen. Thus, upstream genes in the PDH operon were not selected for further characterization in this study, as they displayed general growth defects. Additionally, LpdA is also reported to be involved in the production of staphylococcal protein A (SpA), a surface protein and a critical virulence factor for airway infection: SpA is completely abrogated in JE2 *lpdA*::Tn mutant (NE1610) but not in the mutants defective in other three catalytic polypeptides of PDH (19). Future research to decipher the molecular connection between LpdA and SpA and experiments to determine the role of SpA in MRSA uropathogenesis is warranted. Importantly, it must be noted that exposure to HU significantly downregulates the transcription of *spa* (SAUSA300_0113) (16).

Overall, our results dictate the importance of the TCA cycle, mannitol metabolism, and pyruvate oxidation for the fitness of MRSA in the UT environment. The continual emergence of antibiotic-resistant strains resulting from the acquisition of mobile genetic elements remains the major challenge in the successful treatment of MRSA infections. A better understanding of the metabolic pathways utilized in the host UT for survival and proliferation may help us develop novel therapeutics against MRSA UTI.

**TABLE 2** Strains used in this study

| Strain | Description | Reference(s) or source |
|---|---|---|
| JE2 | Wild-type background strain in which the NTML was generated | 30, 38 |
| NTML | Collection of 1920 transposon mutants in the JE2 background | 30, 38 |
| JE2 *sucD*::tn | Transposon mutant in *sucD* | This study |
| JE2 *fumC*::tn | Transposon mutant in *fumC* | This study |
| JE2 *mltD*::tn | Transposon mutant in *mltD* | This study |
| JE2 *lpdA*::tn | Transposon mutant in *lpdA* | This study |
| MRSA 1369 | Urinary tract clinical isolate | 12 |
| MRSA 1369 *sucD* | Transposon mutant in *sucD* | This study |
| MRSA 1369 *fumC* | Transposon mutant in *fumC* | This study |
| MRSA 1369 *mltD* | Transposon mutant in *mltD* | This study |
| MRSA 1369 *lpdA* | Transposon mutant in *lpdA* | This study |

## MATERIALS AND METHODS

**Bacterial strains, HU, and reagents.** The WT MRSA strain JE2 and the Nebraska transposon mutant library (NTML), which consists of 1,920 mutant strains generated by the insertion of mariner transposon *bursa aurealis* in the genome of MRSA JE2, were provided by the Network on Antimicrobial Resistance in *Staphylococcus aureus* (NARSA, for distribution through BEI Resources, NIAID, NIH: NTML Screening Array, NR-48501) (30, 38). Previously published phage transduction protocols were used to transduce the MRSA 1369 and WT JE2 strains with the transposon mutants from the NTML JE2 background strain that displayed growth defects in HU but not rich brain heart infusion (BHI) medium (see "Generation of mutants" below) (12). Various mutant and WT strains used in this study are listed in Table 2. For simplification, the transposon mutants are referred to by the gene name in the manuscript. The WT and mutant strains were cultured overnight at 37°C with shaking at 200 rpm in BHI broth. Overnight cultures of bacterial strains were centrifuged, washed once in sterile Dulbecco's phosphate-buffered saline (d-PBS) at room temperature, resuspended in d-PBS, and used for experimentation.

HU collected from healthy female volunteers between the ages 18 and 45 years (protocol IRB-22-054-BIOL, approved by the institutional review board [IRB], University of Louisiana at Lafayette) was filter sterilized through a 0.22-$\mu$m filter, aliquoted, and stored at −20°C. At the time of experiment, urine aliquots from at least 3 separate donors were pooled.

**Screening of NTML and WT strains for growth in HU.** To identify mutants that display growth defects in HU but not rich media, all 1,920 NTML mutants, WT JE2, and WT MRSA 1369 were screened *in vitro* for growth in HU and nutrient-rich BHI. Briefly, overnight cultures of each NTML mutant library strain were prepared by inoculating a single colony from a BHI agar plate into BHI broth and incubating it with shaking at 200 rpm at 37°C for 18 h. The culture was diluted 1:100 in either BHI or HU, in triplicate, in a 96-well plate and grown with shaking at 37°C and repeated at least twice. The $OD_{600}$ was measured (BioTek Epoch plate reader) at specified time points for 18 to 24 h. Mutants with a significantly lower $OD_{600}$ than WT JE2 after overnight growth in BHI were excluded from further analyses. Mutants with no difference in $OD_{600}$ from JE2 WT after overnight growth in BHI but significantly lower $OD_{600}$ than WT JE2 after overnight growth in HU were considered defective for growth in human urine. Additionally, growth ratios ($OD_{600}$ mutant/$OD_{600}$ WT) of <0.5 at various time points were also used to assess fitness in HU over time.

**Generation of mutants in the MRSA 1369 background.** To further characterize the TCA cycle mutants (*fumC*, *sucD*), pyruvate oxidation mutant (*lpdA*), and mannitol metabolism mutant (*mltD*), standard phage transduction protocols were used to generate these mutants in the JE2 WT and MRSA 1369 background strains (12). Briefly, phage 11 was used to create lysates of each NTML mutant in *fumC*, *sucD*, *lpdA*, and *mltD* (Table 2). The lysates were then used to transduce MRSA 1369 or the JE2 WT strain that did not undergo transposon mutagenesis. PCR was used to confirm the successful transduction of each mutant.

**Growth kinetics of WT and mutants.** Overnight cultures of MRSA 1369 and mutant strains were pelleted and washed in d-PBS. Bacteria were then inoculated 1:1,000 in pooled HU or BHI and incubated at 37°C without shaking for 24 h. At 0-, 2-, 4-, 6-, 8-, and 24-h time points, a 10-$\mu$L sample was dilution plated to determine CFU/mL on tryptic soy agar (TSA) or TSA containing 10 $\mu$g/mL of erythromycin for transposon mutants where appropriate. For each strain, the doubling time (DT) was calculated as

$$\text{DT} = \left[\text{duration}\left(\log_2\right)\right] / \left[\log(\text{CFU/mL at 4 h}) - \log(\text{CFU/mL at 2 h})\right]$$

For supplementation experiments, we compared the growth curves of WT and *lpdA* mutant in plain HU or HU supplemented with 5 mg/mL glucose or HU supplemented with 5 mM citrate.

***In vitro* competition in HU.** Mid-log-phase cultures of MRSA 1369 WT and mutant strains were washed in d-PBS and inoculated in pooled HU at a 1:1 ratio ($2 \times 10^6$ CFU/mL each for the WT and the mutant). The WT and mutant CFU at 0 h were differentially enumerated by dilution plating the inoculum on TSA and TSA containing 10 $\mu$g/mL erythromycin for total and mutant CFU/mL, respectively. After incubation at 37°C, samples were taken at 4 and 24 h to determine the number of WT and mutant CFU recovered by dilution plating on TSA and TSA containing 10 $\mu$g/mL erythromycin. For 0 (inoculum), 4, and 24 h, WT CFU/mL = total CFU/mL − mutant CFU/mL. The competitive index (CI) for each mutant strain was calculated at 4-h and 24-h time points as follows:

$$\text{CI} = (\text{WT recovered/mutant recovered})/(\text{WT inoculum/mutant inoculum})$$

**Mouse model of CAUTI.** As approved by the Institutional Animal Care and Use Committee (IACUC) at the University of Louisiana at Lafayette (protocol number 2020-8717-025), we experimentally induced CAUTI in 8- to 10-week-old female C57BL/6 mice as described previously (12, 39). In brief, the overnight cultures of MRSA 1369 WT and select mutant strains were inoculated 1:10 in fresh BHI medium and incubated at 37°C with shaking at 200 rpm. Mid-log-phase ($OD_{600} = 0.6$) cultures were then pelleted, washed, and resuspended in d-PBS. An inoculum of 50 $\mu$L (equivalent to $\sim$5 $\times$ $10^7$ CFU/mL) was inoculated immediately after implantation of a 4- to 5-mm piece of silicone tubing (catheter) via transurethral insertion into the bladder of anesthetized mice. The catheter implant remained in the bladder during infection. For single infections, MRSA 1369 WT and mutant strains were inoculated separately into the urinary bladders of anesthetized mice. Mice were sacrificed at 24 hpi, and spleen, kidneys, bladder, and catheter implants were aseptically and sequentially harvested. The CFU were determined by dilution plating the homogenates of bladder, combined halves of bisected left and right kidneys, and spleen. The catheter implant recovered from the bladder was vortexed in 1 mL sterile d-PBS for 1 min to recover bacteria, which were then enumerated by dilution plating. Statistical significance between the organ burdens from WT- and mutant-infected mice was determined using the Mann-Whitney U test.

For *in vivo* competition experiments, mid-log-phase cultures of MRSA WT and mutant were mixed 1:1 to obtain 5 $\times$ $10^7$ CFU/mL total inoculum resuspended in 50 $\mu$L sterile d-PBS. Anesthetized female C57BL/6 mice were then inoculated via transurethral catheterization as described above. The inoculum was dilution plated on plain TSA plates to enumerate total (WT plus mutant) inoculum CFU/mL and on TSA plus 10 $\mu$g/mL erythromycin to enumerate mutant inoculum CFU/mL. After overnight incubation at 37°C, colonies were counted. WT inoculum was equal to total inoculum minus mutant inoculum. At 24 hpi, organs and catheters from infected mice were harvested and processed as described above. These were dilution plated either on TSA for total CFU/mL recovered or on TSA plus erythromycin for mutant CFU/mL recovered. WT recovered equaled total recovered minus mutant recovered. CI was calculated as

$$\text{CI} = (\text{WT recovered/mutant recovered})/(\text{WT inoculum/mutant inoculum})$$

A CI of 1 denotes that both WT and mutant strains are equally competitive in the mouse UT, a CI of <1 represents that the mutant has a competitive advantage over WT for colonizing the mouse UT, and a CI of >1 signifies that the WT has a competitive advantage over the mutant.

**Hydrogen peroxide killing assay.** Overnight cultures of MRSA 1369 WT and mutant strains in BHI were centrifuged, washed in d-PBS, and resuspended at 1:100 dilution in either fresh BHI medium or HU supplemented with 15 mM $H_2O_2$. Inoculum CFU were determined by dilution plating. Cultures were incubated at 37°C with 200 rpm shaking, and bacterial CFU were determined by dilution plating at 4 h.

$$\% \text{ killing} = \left[(\text{initial CFUs} - \text{CFUs at 4 h})/(\text{initial CFUs})\right] \times 100$$

**Cell surface hydrophobicity assay.** Cell surface hydrophobicity was determined using microbial adhesion to hydrocarbon (MATH) assay (16) where mid-log-phase cultures of MRSA 1369 WT and mutant strains ($OD_{600} = 0.6$ in BHI) were centrifuged, washed in d-PBS, and exposed to HU for 2 h. MRSA strains in HU were centrifuged, washed, and resuspended in sterile d-PBS to an $OD_{600}$ of 0.5 and plated for inoculum CFU ($C_i$) enumeration. One milliliter of the bacterial suspension was mixed with 125 $\mu$L hexadecane, vortexed for 1 min, and incubated at room temperature for 30 min. The CFU in the aqueous phase ($C_{aq}$) were then enumerated by dilution plating using the equation % hydrophobicity = $[(C_i - C_{aq})/(C_i)] \times 100$.

**Whole-blood killing assay.** The mid-log-phase cultures of MRSA 1369 WT and mutant strains ($OD_{600}$ of 0.6 in BHI) were centrifuged, washed in d-PBS, and exposed to HU for 2 h. Next, 200 $\mu$L of hirudin-anticoagulated human blood was mixed with 2 $\times$ $10^6$ CFU of the mutant or WT MRSA strains in a 96-well plate. After incubation at 37°C and continuous shaking, samples were extracted at 1 h or 4 h to enumerate surviving bacteria by dilution plating for CFU.

$$\% \text{ killing} = \left[(\text{inoculum CFUs} - \text{CFUs at a specified time})/(\text{inoculum CFUs})\right] \times 100$$

**Statistical analysis.** Statistical tests were performed using Prism 9.0 (https://www.graphpad.com/). Data from multiple biological replicates with two or more technical replicates and two or more biological replicates for each experiment were pooled. Error bars in the doubling time figures represent standard error of the mean. Competitive indices for each mutant strain compared to WT *in vitro* or *in vivo* were analyzed using one-sample *t* test and Wilcoxon test against a theoretical median of 1. For an *in vivo* competition experiment, the CFU per milliliter of WT and mutant strains in each mouse were compared using Wilcoxon matched-pair signed-rank test. The growth, doubling time, percent hydrophobicity, percent killing by $H_2O_2$, and percent killing by human blood for each mutant were compared with that for the WT using unpaired *t* test. The organ burden data from single-infection experiments were compared using Mann-Whitney test. Dissemination rates were compared using Bernard's test. Data were considered statistically significant if *P* values were ≤0.05.

## SUPPLEMENTAL MATERIAL

Supplemental material is available online only.
**SUPPLEMENTAL FILE 1**, TIF file, 7.4 MB.
**SUPPLEMENTAL FILE 2**, TIF file, 9.7 MB.

## ACKNOWLEDGMENTS

We thank BEI Resources for providing the Nebraska Transposon Mutant Library (NR-48501).

This work was supported by grants R21 AI165939 (to R.K.) from the NIH National Institute of Allergy and Infectious Diseases and K01-DK128381-01A1 (to J.N.W.) and R01-DK051406 (to C.L.P.O. and S.J.H.) from the NIH National Institute of Diabetes and Digestive and Kidney Diseases.

We declare no conflicts of interest.

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
