## [Reviewer comments · Microbiology Spectrum]

Microbiology Spectrum

Defining the roles of pyruvate oxidation, TCA cycle, and mannitol metabolism in methicillin resistant *Staphylococcus aureus* catheter-associated urinary tract infection

Ritwij Kulkarni, Santosh Paudel, Sarah Guedry, Chloe Obernuefemann, Scott Hultgren, and Jennifer Walker

Corresponding Author(s): Ritwij Kulkarni, University of Louisiana at Lafayette

Review Timeline:

Submission Date:	December 31, 2022
Editorial Decision:	April 10, 2023
Revision Received:	May 1, 2023
Editorial Decision:	May 29, 2023
Revision Received:	June 5, 2023
Accepted:	June 7, 2023

Editor: Cezar Khursigara

Reviewer(s): The reviewers have opted to remain anonymous.

Transaction Report:

DOI: <https://doi.org/10.1128/spectrum.05365-22>

April 10, 2023

Dr. Ritwij Kulkarni
University of Louisiana at Lafayette
Biology
410, E. St Mary Blvd
Billaeud Hall, Room 108
Lafayette, Louisiana 70504

Re: Spectrum05365-22 (Defining the roles of pyruvate oxidation, TCA cycle, and mannitol metabolism in methicillin resistant *Staphylococcus aureus* catheter-associated urinary tract infection)

Dear Dr. Ritwij Kulkarni:

Two experts have reviewed your manuscript and agree that this is high-quality work. Both suggest changes to the text to align the results with your conclusions better. Please address each comment when submitting a revised manuscript.

Thank you for submitting your manuscript to Microbiology Spectrum. As you will see your paper is very close to acceptance. Please modify the manuscript along the lines I have recommended. As these revisions are quite minor, I expect that you should be able to turn in the revised paper in less than 30 days, if not sooner. If your manuscript was reviewed, you will find the reviewers' comments below.

When submitting the revised version of your paper, please provide (1) point-by-point responses to the issues raised by the reviewers as file type "Response to Reviewers," not in your cover letter, and (2) a PDF file that indicates the changes from the original submission (by highlighting or underlining the changes) as file type "Marked Up Manuscript - For Review Only". Please use this link to submit your revised manuscript. Detailed instructions on submitting your revised paper are below.

Link Not Available

Sincerely,

Cezar Khursigara

Reviewer comments:

Reviewer #1 (Comments for the Author):

This study by Paudel et al explores the contribution of four components of metabolic pathways to *Staphylococcus aureus* growth in urine and pathogenesis in a mouse model of catheter-associated urinary tract infection (CAUTI). This is an important topic, as *S. aureus* represents an understudied but clinically-significant urinary tract pathogen. By screening transposon mutants in the JE2 background, the authors identified mutants in *fumC*, *mtlD*, *lpdA*, and *miaB* as having growth defects in human urine but not rich laboratory medium. All four genes were next mutated in the methicillin-resistant isolate MRSA 1369 and assessed for growth and fitness in urine and during experimental CAUTI. Only the *lpdA* mutant had a growth defect in urine, but all four mutants had fitness defects in urine for at least one timepoint and in at least one organ during CAUTI. The authors further explored the potential contribution of *mtlD* and *lpdA* to ROS sensitivity and serum killing. The manuscript is well-written overall, but there are several places where the conclusions are not fully supported by the data. Specific comments are as follows:

The body of work would be strengthened by confirming that the indicated metabolic pathways are disrupted in each mutant and by ruling out potential off-target effects by complementation, at least for in vitro experiments.

It is unclear if the main effect of each mutation is thought to be the impact on metabolic pathways, or on downstream effect such

as membrane permeability. If the main hypothesis is the impact on metabolism, can the growth defect of the LpdA mutant be complemented by acetyl-coA? Alternatively, if the main impact of the LpdA mutant is thought to be on cell membrane fluidity and subsequent susceptibility to factors such as antimicrobial peptides or complement in blood, it would be beneficial to demonstrate whether the LpdA mutant is more susceptible to membrane-permeable dyes or daptomycin.

It would be useful to include a diagram of the metabolic pathways being explored in this work, denoting which had in vitro and in vivo competitive defects.

Line 273 states that "pyruvate oxidation plays an essential role" in growth and survival in HU, but the LpdA mutant still grows in urine and achieves high CFUs. The expectation for an essential factor is that MRSA would not be able to grow at all without it. Similarly, line 321 states that "LpdA activity is required" for colonization of the lower UT. However, the LpdA mutant still colonized the bladder and catheter of all mice, so it does not appear to be required for colonization. It would be more accurate to state that LpdA "contributes" to colonization. Line 336 references the "essential" roles of LpdA and MtlD, which again is not supported by the data since the LpdA mutant still colonizes and CFUs are not provided for the MtlD mutant. Similarly, line 402 states that the TCA cycle is "required" for MRSA pathogenesis, which overstates the data.

Figure 1A is confusing as presented. For instance, line 256 states that "the LpdA::Tn mutant displayed a severe growth defect at 24h" (which I believe should be corrected to 22 h), but there is only a single, slightly red line for NE1610. By eye, this appears to be similar to the single red line for a mutant that somewhere between NE701 and NE801 at 22 hours, no mutant in this range is mentioned in the text or the RNA-Seq heatmap. I suggest moving the heatmap to supplemental material and displaying growth kinetics of the selected mutants in HU vs BHI for figure 1. I also suggest displaying the actual fold change in expression values, perhaps as a table, rather than a heatmap.

For Figure 4, it would be helpful to include the CFUs for WT and each mutant, in addition to the Cis.

Line 339: Define BCFA on first use

Lines 396-397 state that "sucD and fumC" did not display a competitive disadvantage during growth in human urine, but sucD had a significant defect in Figure 3A.

Reviewer #2 (Comments for the Author):

The article reports on a study that investigated the importance of specific metabolic pathways during Methicillin-resistant Staphylococcus aureus (MRSA) urinary tract infection. The study identified mutations in the TCA cycle (fumC, sucD), mannitol metabolism (mtlD, and pyruvate oxidation (LpdA) that resulted in significant defects in the growth and colonization of MRSA in the urinary tract of mice, suggesting the potential use of these pathways as targets for novel therapeutics. I don't have any major issues with the methodologies and conclusions of this study. However, I do have several suggestions that will hopefully make the manuscript clearer and easier to digest.

- Fig 1A: The authors should indicate which are the 6 mutants that showed significant growth defects in HU but normal growth in BHI. Maybe the zoom-in version of 6 mutants is needed (panel 1B does not need such huge space).
- Fig 1B: the colour scale is difficult to read, the authors should consider providing the logFC value for each gene presented.
- Fig 2: the authors should draw plots in panel B and D in the same style as other figures in this manuscript where individual data points are shown.
- In line 246, the authors should quantify the statement "significant growth defects" - does it mean showing growth ratios < 0.5 in all time points? I see quite a lot of red at 1h and 2h, even 4h in Fig 1A.
- How would the authors interpret the results that out of the 4 mutants that were defective for growth in HU in JE2, only one still showed defective in the 1369 background strain? Are there any differences in the TCA cycle genes between JE2 and 1369?

Preparing Revision Guidelines

Please return the manuscript within 60 days; if you cannot complete the modification within this time period, please contact me. If you do not wish to modify the manuscript and prefer to submit it to another journal, please notify me of your decision immediately so that the manuscript may be formally withdrawn from consideration by Microbiology Spectrum.

We thank the reviewers for the careful review of our manuscript. We have incorporated all of their insightful comments in the revised manuscript. The major changes in the revision include: (i) data from examination of *lpdA* mutant growth in human urine supplemented with glucose or citrate, (ii) clarification on the selection of mutants, and (ii) the addition of a metabolic pathway schematic. The following is the point-by-point response (marked in blue) to the reviewers comments and suggestions. The line numbers are from unmarked version.

Reviewer comments:

Reviewer #1 (Comments for the Author):

This study by Paudel et al explores the contribution of four components of metabolic pathways to *Staphylococcus aureus* growth in urine and pathogenesis in a mouse model of catheter-associated urinary tract infection (CAUTI). This is an important topic, as *S. aureus* represents an understudied but clinically-significant urinary tract pathogen. By screening transposon mutants in the JE2 background, the authors identified mutants in *fumC*, *mtlD*, *lpdA*, and *miaB* as having growth defects in human urine but not rich laboratory medium. All four genes were next mutated in the methicillin-resistant isolate MRSA 1369 and assessed for growth and fitness in urine and during experimental CAUTI. Only the *lpdA* mutant had a growth defect in urine, but all four mutants had fitness defects in urine for at least one timepoint and in at least one organ during CAUTI. The authors further explored the potential contribution of *mtlD* and *lpdA* to ROS sensitivity and serum killing. The manuscript is well-written overall, but there are several places where the conclusions are not fully supported by the data. Specific comments are as follows:

1.1) The body of work would be strengthened by confirming that the indicated metabolic pathways are disrupted in each mutant and by ruling out potential off-target effects by complementation, at least for in vitro experiments.

From our previous experience, complementation of knockout mutants by a plasmid-borne copy of the mutated gene is possible but not always successful in *Staphylococcus aureus* strains. Hence, as an alternative to complementation, we ruled out potential polar effects of mutants by monitoring growth of *lpdA* mutant in human urine supplemented with various downstream (citrate) or alternative (glucose) carbon sources. As shown in supplemental figure S2, *lpdA* showed more robust growth in HU supplemented with either glucose or citrate at 24 h time point.

1. 2) It is unclear if the main effect of each mutation is thought to be the impact on metabolic pathways, or on downstream effect such as membrane permeability. If the main hypothesis is the impact on metabolism, can the growth defect of the *LpdA* mutant be complemented by acetyl-coA? Alternatively, if the main impact of the *lpdA* mutant is thought to be on cell membrane fluidity and subsequent susceptibility to factors such as antimicrobial peptides or complement in blood, it would be beneficial to demonstrate whether the *lpdA* mutant is more susceptible to membrane-permeable dyes or daptomycin.

We observed that *lpdA* mutant growth is improved by the addition of glucose as well as citrate as explained earlier. At 30 h, compared to growth in plain human urine we observed higher *lpdA* CFU/ml in HU+ glucose (not significant by t test) and in HU+citrate ($P=0.054$, t test). These results are presented in Fig S2. In addition, we have revised narrative (lines 485—488) to mention that *LpdA* mutation affects metabolism as well as membrane permeability of MRSA both of which may in turn affect the urinary pathogenesis of *lpdA* mutants.

1.3) It would be useful to include a diagram of the metabolic pathways being explored in this work, denoting which had *in vitro* and *in vivo* competitive defects.

In the revised Fig 1, we have presented a schematic of metabolic pathways (glycolysis and TCA cycle) and have indicated specific metabolic steps catalyzed by MtlD, LpdA, FumC, and sucD.

1.4) Line 273 states that "pyruvate oxidation plays an essential role" in growth and survival in HU, but the *lpdA* mutant still grows in urine and achieves high CFUs. The expectation for an essential factor is that MRSA would not be able to grow at all without it. Similarly, line 321 states that "LpdA activity is required" for colonization of the lower UT. However, the *LpdA* mutant still colonized the bladder and catheter of all mice, so it does not appear to be required for colonization. It would be more accurate to state that LpdA "contributes" to colonization. Line 336 references the "essential" roles of LpdA and MtlD, which again is not supported by the data since the *LpdA* mutant still colonizes and CFUs are not provided for the MtlD mutant. Similarly, line 402 states that the TCA cycle is "required" for MRSA pathogenesis, which overstates the data.

Thank you for the suggestion. We have revised the manuscript to state that LpdA and MtlD are *neither essential nor required* but are important for MRSA uropathogenesis. This change is also reflected in the revised running title: "LpdA and MtlD are required for MRSA uropathogenesis".

1.5) Figure 1A is confusing as presented. For instance, line 256 states that "the *lpdA::Tn* mutant displayed a severe growth defect at 24h" (which I believe should be corrected to 22 h), but there is only a single, slightly red line for NE1610. By eye, this appears to be similar to the single red line for a mutant that somewhere between NE701 and NE801 at 22 hours, no mutant in this range is mentioned in the text or the RNA-Seq heatmap. I suggest moving the heatmap to supplemental material and displaying growth kinetics of the selected mutants in HU vs BHI for figure 1. I also suggest displaying the actual fold change in expression values, perhaps as a table, rather than a heatmap.

Thank you for these suggestions. We have moved the original Fig 1A) to supplemental figure S1; the original Fig 1B) is presented as Table 2. In addition, we have significantly edited the language to avoid confusion about selection of mutants for this study. The revised narrative (Lines 257—273) explains that the four mutants were selected for this study because they were defective for growth in human urine but grew normally in nutrient rich BHI. Three of the four genes (*sucD*, *fumC*, and *mtlD*) were also upregulated when exposed *in vitro* to HU for 2h. The revision also explains that despite showing a growth ratio <0.5 at 24 h, NE751 was not selected for this study because it encodes a hypothetical protein and because it was defective for growth in nutrient rich BHI.

1.6) For Figure 4, it would be helpful to include the CFUs for WT and each mutant, in addition to the *Cis*.

The revised figure for *in vivo* competition shows both competitive indices and raw CFU/ml data.

1.7) Line 339: Define BCFA on first use

This has been revised.

1.8) Lines 396-397 state that "*sucD* and *fumC*" did not display a competitive disadvantage during growth in human urine, but *sucD* had a significant defect in Figure 3A.

This has been revised.

Reviewer #2 (Comments for the Author):

The article reports on a study that investigated the importance of specific metabolic pathways during Methicillin-resistant *Staphylococcus aureus* (MRSA) urinary tract infection. The study identified mutations in the TCA cycle (*fumC*, *sucD*), mannitol metabolism (*mtlD*), and pyruvate oxidation (*lpdA*) that resulted in significant defects in the growth and colonization of MRSA in the urinary tract of mice, suggesting the potential use of these pathways as targets for novel therapeutics. I don't have any major issues with the methodologies and conclusions of this study. However, I do have several suggestions that will hopefully make the manuscript clearer and easier to digest.

2.1) Fig 1A: The authors should indicate which are the 6 mutants that showed significant growth defects in HU but normal growth in BHI. Maybe the zoom-in version of 6 mutants is needed (panel 1B does not need such huge space).

Thank you for this suggestion. As mentioned, in a response to 1.5), we have provided log₂FC values for the four genes examined in this manuscript.

2.2) Fig 1B: the colour scale is difficult to read, the authors should consider providing the log₂FC value for each gene presented.

Thank you for this suggestion. As mentioned, in a response to 1.5), we have provided log₂FC values for the four genes examined in this manuscript.

2.3) Fig 2: the authors should draw plots in panel B and D in the same style as other figures in this manuscript where individual data points are shown.

Thank you for this suggestion. We have revised Fig 2B and D to show individual doubling times as scatter plot with histogram representing the average.

2.4) In line 246, the authors should quantify the statement "significant growth defects" - does it mean showing growth ratios < 0.5 in all time points? I see quite a lot of red at 1h and 2h, even 4h in Fig 1A.

We admit that "significant growth defects" was not adequately defined in the original submission. A similar concern was also raised by the reviewer 1 (please see response to 1.5). We selected *sucD*, *fumC*, and *mtlD* mutants because they showed growth ratio < 0.5 at 2 and 4 h in human urine and because their growth in BHI was not affected. These three genes were also significantly upregulated in MRSA 1369 exposed to HU for 2 h. We selected *lpdA* mutants because it was significantly defective for growth in human urine at 22 h. Moreover, *sucD*, *fumC*, *mtlD*, and *lpdA* encode enzymes in the central carbon metabolism pathways, which is the theme for this project. The revised narrative can be found on lines 257—273.

2.5) How would the authors interpret the results that out of the 4 mutants that were defective for growth in HU in JE2, only one still showed defective in the 1369 background strain? Are there any differences in the TCA cycle genes between JE2 and 1369?

Thank you for this suggestion. We address the differences in *in vitro* HU growth between MRSA JE2, a skin isolate, and MRSA 1369, a urinary isolate, on lines 418—432.

May 29, 2023

Dr. Ritwij Kulkarni
University of Louisiana at Lafayette
Biology
410, E. St Mary Blvd
Billaeud Hall, Room 108
Lafayette, Louisiana 70504

Re: Spectrum05365-22R1 (Defining the roles of pyruvate oxidation, TCA cycle, and mannitol metabolism in methicillin resistant *Staphylococcus aureus* catheter-associated urinary tract infection)

Dear Dr. Ritwij Kulkarni:

Thank you for submitting your manuscript to Microbiology Spectrum. As you will see your paper is very close to acceptance. Please modify the manuscript along the lines I have recommended. As these revisions are quite minor, I expect that you should be able to turn in the revised paper in less than 30 days, if not sooner. If your manuscript was reviewed, you will find the reviewers' comments below.

When submitting the revised version of your paper, please provide (1) point-by-point responses to the issues raised by the reviewers as file type "Response to Reviewers," not in your cover letter, and (2) a PDF file that indicates the changes from the original submission (by highlighting or underlining the changes) as file type "Marked Up Manuscript - For Review Only". Please use this link to submit your revised manuscript. Detailed instructions on submitting your revised paper are below.

Link Not Available

Sincerely,

Cezar Khursigara

Reviewer comments:

Reviewer #1 (Comments for the Author):

The authors have addressed all prior reviewer comments. I only note one concern regarding the conclusions drawn about the lpdA glucose and citrate experiments.

Lines 297-298 state that growth of the lpdA mutant was delayed in HU+glucose, but growth appears identical to that of the lpdA mutant in HU without glucose in S2A and S2B until the 30h time point. The text here also states that growth of the lpdA mutant was partially restored in HU+citrate, but the growth curves in S2C are largely identical except for what might be a very small difference at the 24 and 30 time points (hard to tell as there are no error bars for the 30h time point, and S2D shows a P value of 0.054 for the 24 hour time point). How do the CFUs for the lpdA+glucose compare to lpdA alone and wt+glucose at the 30 h time point? This comparison would be needed to support the stated conclusion that "growth of lpdA mutant in HU can be rescued to some extent by supplementation with glucose...or citrate"

Reviewer #2 (Comments for the Author):

The authors have adequately addressed all of my concerns.

Preparing Revision Guidelines

Please return the manuscript within 60 days; if you cannot complete the modification within this time period, please contact me. If you do not wish to modify the manuscript and prefer to submit it to another journal, please notify me of your decision immediately so that the manuscript may be formally withdrawn from consideration by Microbiology Spectrum.

We thank the reviewers for carefully reviewing our resubmission and suggesting edits. We have incorporated the reviewers' suggestions: (i) Supplementary figure #2 (examination of *lpdA* mutant growth in human urine supplemented with glucose or citrate) is edited to include statistical comparisons between different groups, (ii) the sentences describing S2 are edited for clarity. The following is the point-by-point response (marked in blue) to the reviewers comments and suggestions. The line numbers are from the unmarked version.

Reviewer comments:

Reviewer #1 (Comments for the Author):

The authors have addressed all prior reviewer comments. I only note one concern regarding the conclusions drawn about the *lpdA* glucose and citrate experiments.

Lines 297-298 state that growth of the *lpdA* mutant was delayed in HU+glucose, but growth appears identical to that of the *lpdA* mutant in HU without glucose in S2A and S2B until the 30h time point. The text here also states that growth of the *lpdA* mutant was partially restored in HU+citrate, but the growth curves in S2C are largely identical except for what might be a very small difference at the 24 and 30 time points (hard to tell as there are no error bars for the 30h time point, and S2D shows a P value of 0.054 for the 24 hour time point). How do the CFUs for the *lpdA*+glucose compare to *lpdA* alone and wt+glucose at the 30 h time point? This comparison would be needed to support the stated conclusion that "growth of *lpdA* mutant in HU can be rescued to some extent by supplementation with glucose...or citrate"

We apologize for the mistake in the figure legend for Fig S2. Panels B and D show data for 30 h time point. This has been corrected.

We have edited Figure S2 as follows- 1) the y axis in edited panels B and D start from 100,000 making the differences between histograms discernible, 2) asterisks are shown for comparisons where the difference between groups is statistically significant, 3) the duplicate data points for each group are shown in Fig S2B and Fig S2D.

The edited description of Fig S2 reads as follows (Lines 294-303):

"Next, we compared the growth of the WT and *lpdA* mutant strains in HU supplemented with glucose or with citrate (Fig S2). The *lpdA* mutant growth in HU+glucose (Fig S2A) or in HU+citrate (Fig S2C) was affected up to 24 h time point similar to what was observed in HU alone. However, at 30 h time point, the *lpdA* mutant showed ~3-fold higher CFU/ml in HU+glucose (3.2×10^7 CFU/ml, not significant, Fig S2B) and ~2-fold higher CFU/ml in HU+citrate (2.1×10^7 CFU/ml, $P=0.054$, unpaired t test, Fig S2D) compared to HU alone (9.4×10^6 CFU/ml). These results suggest that the lower growth of *lpdA* mutant in HU can be rescued modestly but statistically insignificantly by supplementation with glucose, an alternative carbon source, or citrate which is downstream of the metabolic step catalyzed by *LpdA*."

Reviewer #2 (Comments for the Author):

The authors have adequately addressed all of my concerns.

Thank you.

June 7, 2023

Dr. Ritwij Kulkarni
University of Louisiana at Lafayette
Biology
410, E. St Mary Blvd
Billaeud Hall, Room 108
Lafayette, Louisiana 70504

Re: Spectrum05365-22R2 (Defining the roles of pyruvate oxidation, TCA cycle, and mannitol metabolism in methicillin resistant *Staphylococcus aureus* catheter-associated urinary tract infection)

Dear Dr. Ritwij Kulkarni:

Your manuscript has been accepted, and I am forwarding it to the ASM Journals Department for publication. You will be notified when your proofs are ready to be viewed.

Sincerely,

Cezar Khursigara
Editor, Microbiology Spectrum
